# Decoupled Monitoring Method for Strain and Cracks Based on Multilayer Patch Antenna Sensor

**DOI:** 10.3390/s21082766

**Published:** 2021-04-14

**Authors:** Zhiping Liu, Qian Guo, Yuanhao Wang, Biwei Lu

**Affiliations:** 1School of Logistic Engineering, Wuhan University of Technology, Wuhan 430063, China; gq@whut.edu.cn (Q.G.); 263511@whut.edu.cn (Y.W.); lulbw@whut.edu.cn (B.L.); 2Engineer Research Center of Logistic Technology and Equipment, Ministry of Education, Wuhan 430063, China

**Keywords:** multilayer structure, patch antenna sensor, strain identification, crack monitoring, decoupling

## Abstract

As the rapid development of high-end intelligent equipment continues, the design requirements for crack and strain monitoring equipment are increasing daily. In this paper, a decoupled monitoring method for strain and cracks based on a multilayer patch antenna sensor is studied. First, the monitoring principle for strain and crack decoupling is analyzed. Second, the design method for the multilayer patch antenna sensor is studied, and the hierarchical arrangement, patch size, substrate layer thickness, and feeding line structure are designed on the basis of this method. A quarter-wavelength impedance converter is designed to perform impedance matching and optimize the resonant frequency information. The effects of strain and crack propagation on the resonant frequency of the patch antenna are analyzed through simulations, and the decoupled monitoring method for the structural stress state and crack propagation is discussed. Lastly, the feasibility of decoupled monitoring of strain and cracks is verified experimentally. The results of the theoretical analysis, simulations, and experiments show that the proposed patch antenna sensor based on the multilayer structure can realize decoupled monitoring of strain and cracks in the structure, and the sensor has broad application prospects.

## 1. Introduction

Metal structures are used widely in aerospace, mechanical equipment, and civil and marine engineering applications. Because of the poor working conditions for mechanical equipment, stress concentration, fatigue deformation, and crack initiation often occur within the weak links of structures, including complex stress areas and key welds, and they pose a serious threat to safe equipment use. There has been a long-term focus in the mechanical industry and among researchers worldwide on health monitoring research in metal equipment structures to ensure the health and safety of these structures [1,2].

Traditional metal structure health monitoring methods mainly use either a single sensor or a combination of sensors (including metal strain gauges, piezoelectric acceleration sensors [3,4], various optical fiber sensors [5,6] and acoustic emission sensors [7,8,9]) for strain testing, vibration testing, crack identification, and crack growth monitoring of these structures. To perform structural health monitoring under complex working conditions, the single testing functions of traditional sensors mean that large numbers of sensor nodes are usually arranged to form a sensor network, which can present some problems in practical use, including wiring difficulty, high installation costs, complex structures, and low efficiency. Patch antenna sensors offer the advantages of simple structures, low cost, easy installation, and ease of conformation with complex structures; these sensors can detect structural strain, cracks, and expansion through changes in resonant frequency and, thus, have been widely researched [10].

At present, patch antenna sensors are mainly used to monitor the structural health of single parameters, e.g., cracks, strain, and temperature. Tata et al. first proposed use of rectangular microstrip patch antenna sensors to detect and identify cracks in metal structures [11]. Cracks in the ground plane of the microstrip antenna sensor cause the current to flow around the crack tip and increase the electrical path, thus causing the microstrip antenna sensor’s resonant frequency to change. Qian et al. designed a microstrip patch antenna strain sensor operating at 5 GHz using the relationship between the resonant frequency and the length of the microstrip patch antenna. The microstrip patch antenna’s resonant frequency decreases linearly in tandem with increases in the applied strain [12]. Sanders studied the effects of temperature on the resonant frequency of these antennas as a function of the relationships among the dielectric constant of the microstrip antenna substrate, the physical size of the radiation patch, and the operating temperature, and the microstrip patch antenna was used as the temperature sensor for their research [13].

In engineering, monitoring of a single parameter is often insufficient to meet structural health monitoring requirements under complex working conditions. When performing local safety evaluations, a comprehensive consideration of multiple parameters is essential and the design and development of a multifunction patch antenna sensor is urgently required in engineering. Cho et al. introduced a passive (battery-free) wireless frequency-doubled antenna sensor for strain and crack detection. Their sensor can detect small strain changes and small crack propagation but can only be used to perform separate crack and strain tests [14]. Tchafa et al. developed an inverse calculation method to determine the variations in antenna resonant frequencies with strain and temperature changes and attempted to analyze the magnitudes of these strains and temperatures from a calculation perspective [15].

When patch antenna sensors are applied in structural health monitoring under complex working conditions, the problem of multiple physical field coupling and decoupling must be faced first. Multi-physical field coupling studies of patch antennas are mainly performed to explore the relationships among the temperature field (e.g., heating of active devices), structural field deformation (deformation caused by heating), and the antenna’s electromagnetic field, which can then be used to aid in accurate design of the antenna parameters or improvement of the antenna gain and other radiation performance parameters. Ren-xi et al. proposed a joint electromagnetic–thermal–stress simulation method to predict the radiation and other working performance characteristics of patch antennas under multi-physical coupling field conditions [16]. Wang et al. introduced a new type of radiofrequency heating device composed of an antenna array, and the device’s temperature distribution was calculated as a function of the coupling relationship between the electromagnetic field and the temperature field [17]. These studies essentially form the basic research on the antenna multi-physical field and were used to improve the radiation performance of the patch antenna. The research into multi-physical field coupling involved has referential significance for the research into multi-physical field decoupling for the patch antenna sensor. Through appropriate design of the antenna size, structure, and substrate material, multifrequency characteristics can be obtained from multilayer patch antennas that are mainly used to improve the antenna bandwidth and gain and optimize the antenna radiation performance. Sun et al. introduced a broadband cavity-backed proximity-coupled stacked microstrip antenna. As a result of the interaction of the two stacked patches, two associated resonances were produced [18]. Zhang et al. introduced a novel circularly polarized microstrip antenna with a broadband superposed structure that broadened the antenna impedance bandwidth and circularly polarized bandwidth and could continue to be superposed to expand the bandwidth further [19]. The multifrequency characteristic of the multilayer patch antenna was used in the above researches to perform frequency superposition. From another perspective, this multifrequency characteristic can also be used to perform multi-physical field decoupling research.

In this paper, with the aim of addressing the problem of the single monitoring parameters of patch antenna sensors, a multilayer structure is applied innovatively to a patch antenna sensor. Decoupled monitoring of strain and cracks is realized using the multi-cavity model. The method for multilayer patch antenna sensor design is then studied on this basis. The patch resonance information is optimized by appropriate design of the feeding line size and structure. Lastly, the simultaneous monitoring of the strain and crack information is realized.

## 2. Principle of Strain and Crack Decoupled Monitoring for Multilayer Antenna Sensor

The basic patch antenna sensor structure, consisting of a dielectric substrate layer, a patch coated on one side of the substrate, and a ground plane on the other side of the substrate, is shown in Figure 1. In the figure, both the patch and the ground plate are perfect electric conductors. *L* represents the patch length, *W* represents the patch width, and *H* represents the substrate thickness. By feeding the patch antenna in a specific manner, a resonant cavity can be formed between the rectangular patch made from conductive metal and the ground plane, and electromagnetic waves with a specific resonant frequency are then radiated outward through the gap between the patch and the ground plane.

The resonant frequency of the patch antenna sensor can be calculated using Equation (1), where *c* is the speed of light in a vacuum, 𝑓_r_ is the antenna working frequency, ε_e_ is the effective dielectric constant of the substrate, and ∆*L* is the increment in the electric length caused by the edge effect [20].
(1)$$f_{r}=\frac{c}{2\sqrt{\varepsilon_{e}}}\frac{1}{L+2\;∆L}.$$

The multilayer patch antenna sensor structure is shown in Figure 2. A substrate layer and a patch are added to the basic traditional patch antenna sensor. The structure is loaded along the length and width direction; *F*’ and *F* represent the load, respectively, *L*_1_ and *W*_1_ represent the lower patch length and width, respectively, *L*_2_ and *W*_2_ represent the upper patch length and width, respectively, and *L*_2_’ and *W*_2_’ represent the upper patch length and width deformed, respectively. The resonant frequency calculation for each layer of the patch antenna is then the same as that for a single-layer patch antenna [21,22].

As shown in Figure 3, when the metal structure is loaded, the upper layer of the patch is subjected to stress and deformation through hierarchical stress transmission, and the deformed patch shape is represented by a dotted line. The strains generated along the patch length and width are *ε_x_* and *ε_y_*, respectively, the length is *L*’, the width is *W*’, and the substrate thickness is *H*’. The length and width calculation equations are Equation (2) and Equation (3), respectively, where *γ_p_* represents Poisson’s ratio for the patch.
(2)$$L^{\prime }=(1\;+\;\varepsilon_{x}\;-\;\gamma_{p}\varepsilon_{y})L$$
(3)$$W^{\prime }=(1\;+\;\varepsilon_{y}\;-\;\gamma_{p}\varepsilon_{x})W$$


By combining Equations (2) and (3), the resonant frequency variation of the patch can be obtained as follows:(4)$$∆f_{r}=\frac{c}{2\sqrt{\varepsilon_{e}}}\frac{1}{L^{\prime }+2\;∆L^{\prime }}-\frac{c}{2\sqrt{\varepsilon_{e}}}\frac{1}{L+2\;∆L}.$$

The resonant frequency when affected by a crack is expressed as
(5)$$f_{r}(crack)=\frac{c}{2\sqrt{\varepsilon_{e}}}\frac{1}{(L+∆L_{crack})+2∆L},$$
where $∆L_{crack}$ is the change in the electric length caused by the crack.

As shown in Figure 4, when the lower patch is fed, a radiofrequency electromagnetic field is produced between the patch and the ground plate. The space between the lower patch and the ground plate can be regarded as a cavity bounded by electric conductors (above and below it) and by magnetic walls (to simulate an open circuit) along the perimeter of the patch [23]. When cracks appear on the structure’s surface, the current on the cavity surface bypasses them, causing the lower layer patch resonant frequency to change. When the upper layer is fed, the lower layer is used as the ground plate for the upper layer, which causes the space between the upper and lower layers to form a resonant cavity; at this time, because of the structure, a surface crack would not affect the current distribution on the lower layer surface, i.e., the surface current on the ground plate of the upper resonant cavity is not affected by the crack, thus indicating that the crack has little effect on the resonant frequency of the upper layer patch.

When the structure is loaded, the stress is transferred upward through the hierarchical structure, and the patch is stressed and deformed. The resonant frequency of the patch is then shifted because of the change in size. The magnitude of the resonant frequency offset caused by the crack is of the order of MHz/mm, which is much larger than that caused by the strain, which is of the kHz/με order. Throughout the paper, we use the unit ε to indicate the amount of the strain, and $1\;\mu \varepsilon =10^{-6}\;\varepsilon$. The strain test value of the upper patch can be used to compensate slightly for the resonant frequency offset caused by the stress deformation of the lower patch.
(6)$$f_{r}(crack)=f_{r}^{\prime }(crack)\times \left(1+\varepsilon \times \delta \right),$$
where $f_{r}^{\prime }crack$ is the test resonant frequency for the lower layer patch, and $∆f_{\varepsilon }$ is the resonant frequency offset due to strain.
(7)$$∆f_{\varepsilon }=\varepsilon \times \delta \times R_{\varepsilon },$$
where ε is the strain value measured using the upper patch, δ is the strain transmission efficiency of the upper and lower layers, and $R_{\varepsilon }$ is the strain change rate of the resonant frequency of the lower layer.

The structural cracks are monitored using the lower layer patch, which has little effect on strain monitoring of the upper layer patch. The resonant frequency shift for the lower layer patch caused by the structural stress is much smaller than that caused by the crack, and the lower layer patch crack monitoring information can be compensated for slightly using the test strain from the upper layer patch to realize decoupling of the crack information and the strain information from the structure using the multi-resonator model.

## 3. Design of Multilayer Patch Antenna Sensor

The multilayer patch antenna sensor is used for multifunction parameter monitoring. The design parameters of the multilayer patch antenna sensor are shown in Figure 5. Because of the different test functions of each layer, the size and structure of each patch layer are designed according to their test function needs. The feeding line structure should be designed rationally to ensure a reasonable power distribution for each patch. In addition, the return loss at the resonant frequency should meet the engineering needs, i.e., the return loss at the crest should be no greater than −10 dB.

To ensure high strain testing sensitivity and wide-ranging crack identification, it is necessary to design the patch size to be sufficiently large under the premise of sensitivity for crack identification and to set up a smaller-sized upper patch to test the plane strain. In this study, an external two-channel power divider is used to feed and collect information from the multilayer patches simultaneously, where the patch size of each layer is designed to ensure the required degree of isolation of the resonant frequency.

The lower patch used for crack monitoring is labeled Patch 1, and the upper patch for strain monitoring is labeled Patch 2. An FR4 glass fiber epoxy resin board with good heat and moisture resistance was selected as the substrate. The upper and lower substrate thicknesses were both 1 mm, and the substrate dielectric constant was 4.4. The design parameters of upper and lower patch antennas are given in Table 1.

The patch antenna feeding modes can be mainly divided into the microstrip line, coaxial probe, aperture coupling, and proximity coupling. For convenience in manufacture, a simple structure, and avoidance of damage to the structure, microstrip line feeding is used in this work on the basis of a feasibility analysis of structural damage monitoring. To realize multifunction testing of the sensor based on the multifrequency characteristics of the multilayer patch antenna, the microstrip line is fed at one-fourth of the length of the long side of the patch in this design. Moreover, two resonant modes, $TM_{10}$ and $TM_{01}$, are excited simultaneously by this form of feed, which means that each patch produces two resonant frequencies.

In engineering, the resonant frequency with a peak return loss of less than −10 dB is defined as the effective resonant frequency [24]. Impedance matching of the feeding lines provides an effective way to reduce the return loss. At present, quarter-wavelength impedance converters are widely used for this purpose. On this basis, the feeding line structures are designed optimally in this study.

The design of the multilayer patch feeding lines is optimized via simulations in High Frequency Structure Simulator (HFSS. Ansys Inc., Canonsburg, PA, USA) electromagnetic simulation software. The simulation model is shown in Figure 6. The double-layer substrate material was FR4_epoxy (Kingboard Holdings Co. Ltd., Shatin, Hong Kong), which has a dielectric constant of 4.4, the ground plate material was stainless steel, the plate thickness was 6 mm, and the upper and lower patches were fed directly using a quarter-wavelength impedance converter. The patch and the boundary condition of the floor were set to “perfect electric conductor”, and the feeding line terminal was provided with a 50 Ω excitation port. An air layer was set to contain the entire multilayer patch antenna sensor to simulate the real environment.

Because of the coupling phenomenon that occurs between feeding lines on different layers, it is necessary to reduce the coupling interference to eliminate the effects of the upper and lower feeding lines on the resonant frequencies of the other layers during the structural design process by adjusting the substrate layer thickness. This enables independent optimization of the upper and lower feeding line structures. The substrate thickness is optimized in the HFSS software.

When the substrate thickness of the upper layer is 1 mm, changes in the feeding line structure for each layer have almost no influence on the resonant frequencies of the patches of the other layers, i.e., the two-layer feeding line structure can be optimized independently; the simulation results are as shown in Figure 7.

The diagram of patch antenna feeding lines is shown in Figure 8. According to the principle of the quarter-wavelength impedance converter [25], the lengths of the bc segment and the ef segment were selected to be ne-fourth of the wavelength of the corresponding patch, and the microstrip line width was designed preliminarily using the microstrip line resistance value calculation software to provide a reasonable power distribution. Design parameters of the multilayer patch antenna sensor are given in Table 2.

To ensure that the return loss at the patch antenna working resonant frequency is as small as possible to meet the engineering requirements, the widths of two groups of feeding lines, i.e., segment ab and segment bc, and segment de and segment ef, were simulated and optimized in the HFSS software.

As shown in Figure 9, the simulation results show that, when $W_{ab}$ = 3.5 mm and $W_{bc}$ = 2.5 mm, the return loss at the working resonant frequency of the lower patch reached minimum comprehensively with consideration of the resonant frequencies $f_{10}$ and $f_{01}$. When $W_{de}$ = 2.5 mm and $W_{ef}$ = 1.5 mm, the return loss of the resonant frequency of the upper patch reached minimum.

## 4. Strain and Crack Monitoring of Multilayer Antenna Sensor

The multilayer patch antenna sensor realizes decoupling of the cracks and strain from the structure, and strain and crack monitoring can then be carried out simultaneously, i.e., the lower patch is used for crack monitoring and the upper patch is used for strain identification; in addition, cracks have no effect on the upper patch, and the influence of strain on the lower patch is compensated for slightly using the strain measured via the upper layer.

### 4.1. Simulation Analysis of Strain and Crack Monitoring

In this section, crack growth on the structural surface under a stress field is simulated in the HFSS software and the effects of crack growth and strain on the resonant frequencies of the patches in each layer are analyzed [26,27]. The simulation model diagram is shown in Figure 10.

#### 4.1.1. Crack Monitoring Analysis

As shown in Figure 10, the growth of a penetrating crack with width of 0.5 mm was simulated in a Boolean operation module. The horizontal crack extended from the center of the lower patch to the patch edge with an extension step length of 4 mm and total extension length of 60 mm. The simulation results are shown in Figure 11. Crack propagation in the length direction mainly affected the resonant frequency $f_{01}$ under the $TM_{01}$ mode of the lower layer patch and the resonant frequency of $f_{01}$ was, thus, observed. The vertical crack extended from the center of the lower patch to the patch edge along its width with a step length of 4 mm and total growth length of 40 mm. The simulation results are shown in Figure 11. Crack propagation in the width direction mainly affected the resonant frequency $f_{10}$ under the $TM_{10}$ mode of the lower layer patch and the resonant frequency of $f_{10}$ was, thus, observed. The figure shows that, with increasing crack propagation, the working resonant frequency of the patch antenna decreased parabolically.

#### 4.1.2. Decoupling Identification Analysis of Strain

When a crack propagates on the ground layer under an applied load, the trend for variation of the working resonant frequency of the upper layer patch with changes in strain is observed. Consider the case where the load is applied in the length direction as an example; when the crack on the ground layer expands from 0 mm to 12 mm with a step length of 4 mm, the resonant frequency change in the upper patch is as shown in Table 3, and the relationship between the strain and the resonant frequency is shown in Figure 12.

The table and the change trend diagram above show that the upper layer patch resonant frequency decreased linearly with increasing strain during the crack expansion process from 0 mm to 12 mm. The abnormal data point in the figure may have been caused by the error of the adaptive network in the numerical solution. The strain sensitivities were 2.1327 kHz/με, 2.0909 kHz/με, 2.0727 kHz/με, and 2.1236 kHz/με. The theoretical sensitivity was 2.0378 kHz/με, and the strain sensitivity error values were 0.0949, 0.0531, 0.0349, and 0.0858 kHz/με, respectively. These results show that the crack propagation occurring in the structure has little effect on the strain identification and that there is a small error within the allowable range caused by differences between the simulation models.

### 4.2. Strain Compensation Analysis

When the structure is loaded, the lower patch size also changes. The crack propagation under different strain size conditions was simulated using the loading direction along the patch length as an example. The crack length expanded from 4 mm to 40 mm, and the variations in the patch resonant frequency were compared when the strain sizes were 0, 1‰, 2‰, 3‰, 4‰, and 5‰. The simulation results are shown in Figure 13, and the mean values of the relative resonant frequency offset were calculated under strains of different sizes and without strain, with results as shown in Table 4.

The simulation results show that changes in the strain state only change the working resonant frequency of the lower layer patch and do not affect the resonant frequency variation trend during cracking, i.e., it does not affect the crack monitoring effect of the lower layer patch. The resonant frequency offset caused by strain can be calculated and compensated for using the strain test value from the upper layer patch.

## 5. Experimental and Data Analysis

This section begins by presenting the crack testing and strain identification performances of the patch antenna sensor; then, micro-compensation research based on the crack test results is carried out using the test strain, and the crack and strain decoupling performance of the multilayer patch antenna sensor is verified.

### 5.1. Crack and Strain Decoupling Test

The test platform included the multilayer patch antenna sensor, a coaxial connector (SMA. Tejiate Technology Co. Ltd., Shenzhen, China), a vector network analyzer (Agilent E5061B, 3 kHz–3 GHz. Agilent Technologies Inc., Santa Clara, CA, USA), a universal tensile testing machine (THMELX-2. Zhejiang Tianhuang Science & Technology Industrial Co. Ltd., Hangzhou, China), a strain gauge (BX120-3AA. Beijing Yiyang Strain and Vibration Testing Technology Co. Ltd., Beijing, China), a static strain gauge (XL2101C. Xieli Science & Technology Co. Ltd., Qinhuangdao, China), and an aluminum specimen. The test device is shown in Figure 14. The multilayer patch antenna sensor was bonded to the central area on one side of the aluminum specimen using a strong adhesive and was connected to the vector network analyzer through the coaxial connector, and the frequency domain signals from each layer of the patch antennas were collected using the multiport vector network analyzer. The strain gauge was bonded to the central area on the other side of the specimen and is connected to the static strain gauge. The arrangements on both sides of the aluminum specimen are shown in Figure 15. The sample was preprocessed to simulate a straight penetrating crack at various lengths. The crack length ranged from 0 mm to 60 mm in step lengths of 4 mm to simulate the various stages of crack growth.

To reduce test errors, measurements were taken three times at each crack length level, and the resonant frequency information from the lower crack test patch and the upper strain test patch was measured simultaneously during the crack growth process. To provide further observations of the resonant frequency variations in the upper strain test patch during crack propagation, the rate of resonant frequency change of the upper patch was normalized, and the test results are shown in Figure 16.

The experimental results show that, as the crack expanded along the length direction, the resonant frequency $f_{01}$ of the lower crack test patch under the $TM_{01}$ mode presented a parabolic decline trend, while the resonant frequency $f_{10}$ of the lower crack test patch under the $TM_{10}$ mode basically remained unchanged, which is consistent with the simulation results. The test error at 20 mm crack length may have been caused by the contact problem at the SMA joint during the first test. During crack propagation, the normalized rates of change of the resonant frequency of the upper strain test patch in each mode remained within ±0.04, and the population tended to remain stable. In other words, the working resonant frequency of the upper strain test patch was unaffected, and the small fluctuations were mainly related to manufacturing errors in the sample, the bonding process, and the test environment, while the welding condition of the SMA joint used in signal acquisition also had an influence.

Pretreated specimens with different crack lengths were clamped onto the universal tensile testing machine. The loading program was programmed in the tensile testing machine’s control terminal. To reduce the error from the specimen itself, the same preloading force was applied initially to each specimen, and then a controlled load was applied step-by-step on the testing machine. The load range was 0–12 kN, the step size was 3 kN, and the load state was maintained at each stage for 180 s to allow data to be read out by the static strain gauges and vector network analyzers in as stable a state as possible. The tests were repeated five times. The average values of strain measurements of the strain gauge and resonant frequency measurements of the upper patch were used as the test data for this loading range. The test results are shown in Figure 17.

The experimental results show that the resonant frequency of the patch antenna sensor had a good linear relationship with the structural strain when the structural crack extended from 0 mm to 12 mm in steps of 4 mm. The linearity values were 0.974, 0.987, 0.983, 0.995 and 0.975, and the corresponding strain sensitivities were 0.9835 kHz/µε, 0.9428 kHz/µε, 0.9101 kHz/µε, 1.0243 kHz/µε, and 0.9315 kHz/µε, respectively. A certain strain transfer loss occurred because of the use of adhesive layer bonding in the multilayer structure, which caused the strain test sensitivity to decrease during the test. However, the overall variation trend was consistent with the simulation results, and the strain sensitivity at the different crack lengths was basically the same. In other words, the strain test performance of the upper patch remains unaffected, within the allowable error range, during crack propagation. The strain transfer loss caused by the adhesive layer can be reduced by the integration process of hot pressing in practical engineering application.

### 5.2. Strain Compensation

For this example, the structure was loaded step-by-step along the patch length. With the crack length ranging from 0 mm to 16 mm, the resonant frequency deviation of the lower test patch under the different strain states was tested, and each test was repeated five times. The crack in the length direction was represented by the resonant frequency $f_{01}$ of the lower crack testing patch. The offset of the resonant frequency $f_{01}$ caused by the change in the structural strain at each crack length was read, and the average value from multiple tests was taken for the test data. The test results are shown in Figure 18.

After consideration of the system error for specimens with different crack length, the average values of the slopes of each fitted line were selected to represent the rate of change in the resonant frequency $f_{01}$ of the lower patch of the batch of multilayer antenna sensors during the loading process. This means that, when the structure was loaded along the length direction, the rate of change in the resonant frequency $f_{01}$ of the lower patch according to the structural strain was 0.2008 kHz/µε.

According to Equations (6) and (7), the strain state of the lower crack test patch is obtained from the strain tests performed using the upper strain test patch, which means that the resonant frequency shift in the lower layer patch caused by structural strain can be calculated using the rate of resonant frequency change calibrated using the tests. Therefore, microstrain compensation is performed on the lower patch resonant frequency to obtain the crack test value with increased accuracy.

## 6. Conclusions

In this paper, a decoupled monitoring method for strain and cracks based on use of a multilayer patch antenna sensor was studied. The multilayer structure was applied innovatively to the patch antenna sensor. A multilayer patch antenna sensor was, thus, designed for multiparameter decoupling monitoring. The feasibility of the proposed strain and crack decoupling method was analyzed theoretically, simulated, and verified experimentally; the following conclusions were drawn:According to the design of traditional patch antenna sensors, the multilayer patch antenna sensor design must consider the reasonable power distribution required for each layer. Better results can, thus, be obtained through quantitative optimization of the feeding line structure. The design presented in this study can serve as a reference for future multilayer patch antenna sensor designs.The multilayer structure can be used to isolate the effects of surface cracks on the strain identification patch in the upper layer.The effect of a crack on the resonant frequency of the multilayer patch antenna sensor is on the MHz/mm scale, and the effect of strain on the resonant frequency is on the kHz/με scale. The resonant frequency of the lower layer can be compensated got using the strain test value from the upper layer.The multilayer patch antenna sensor realizes decoupled monitoring of strain and cracks. The relationship between the cracks and the antenna resonant frequency is parabolic, and the relationship between the strain and the antenna resonant frequency is linear.

## Figures and Tables

**Figure 1 sensors-21-02766-f001:**
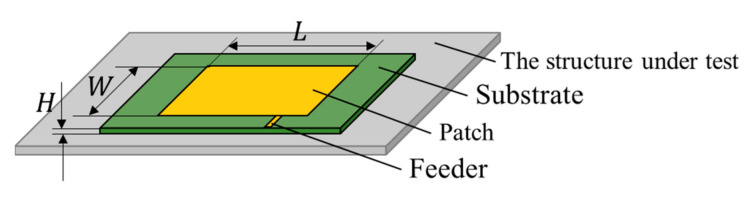
Basic patch antenna sensor structure.

**Figure 2 sensors-21-02766-f002:**
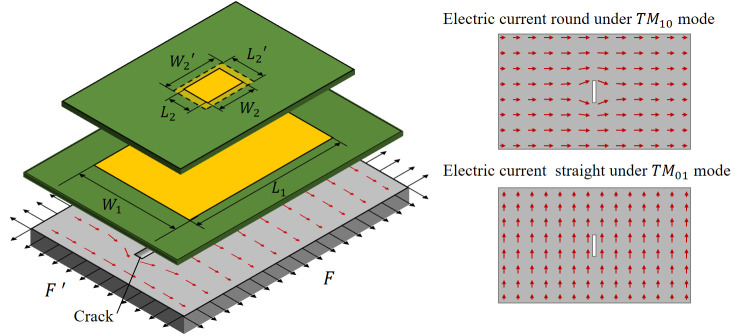
Schematic diagram showing working principle of multilayer patch antenna sensor.

**Figure 5 sensors-21-02766-f005:**
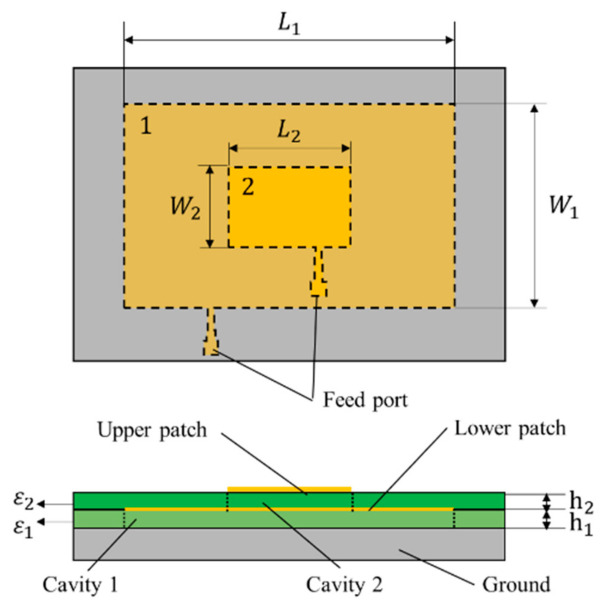
Schematic diagram of multilayer patch antenna sensor.

**Figure 6 sensors-21-02766-f006:**
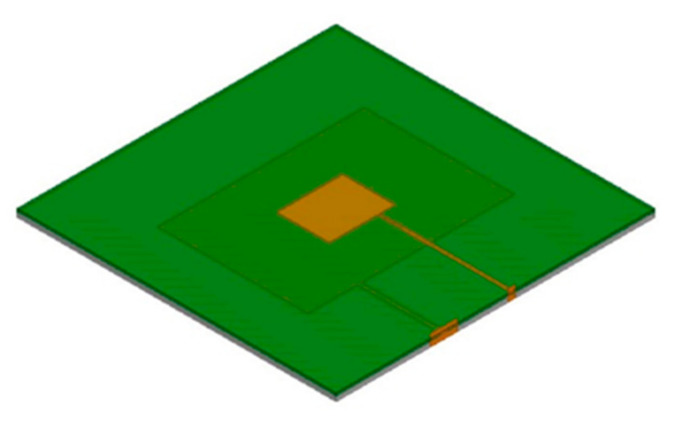
Schematic diagram of the simulation model.

**Figure 7 sensors-21-02766-f007:**
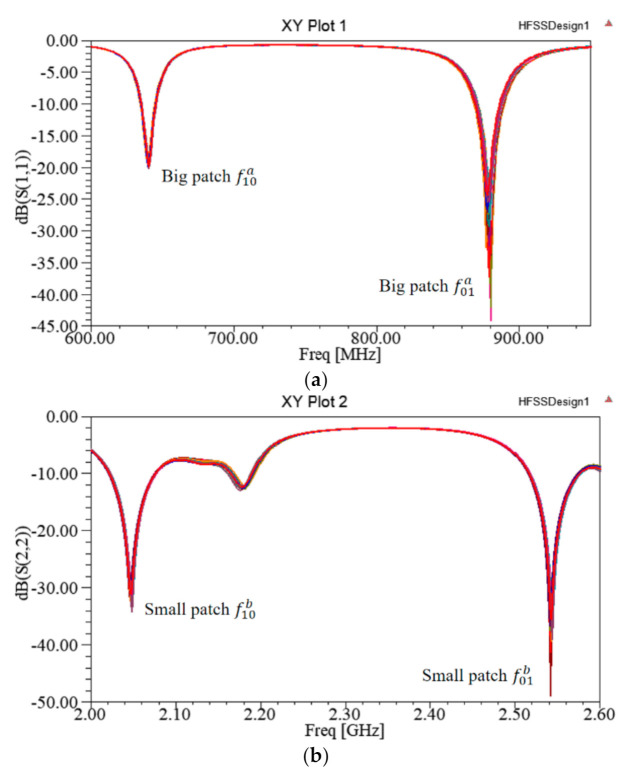
Working resonant frequencies of each layer when the substrate thickness is 1 mm; (**a**) lower patch resonant frequency (with different feeding structure to upper layer); (**b**) upper patch resonant frequency (with different feeding structure to lower layer).

**Figure 8 sensors-21-02766-f008:**
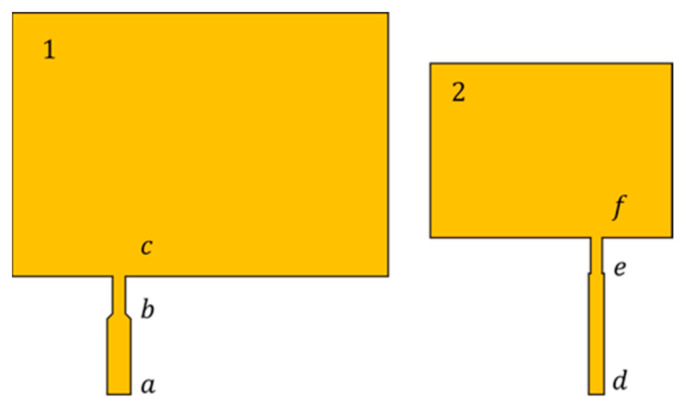
Schematic diagram of the feeding line design.

**Figure 9 sensors-21-02766-f009:**
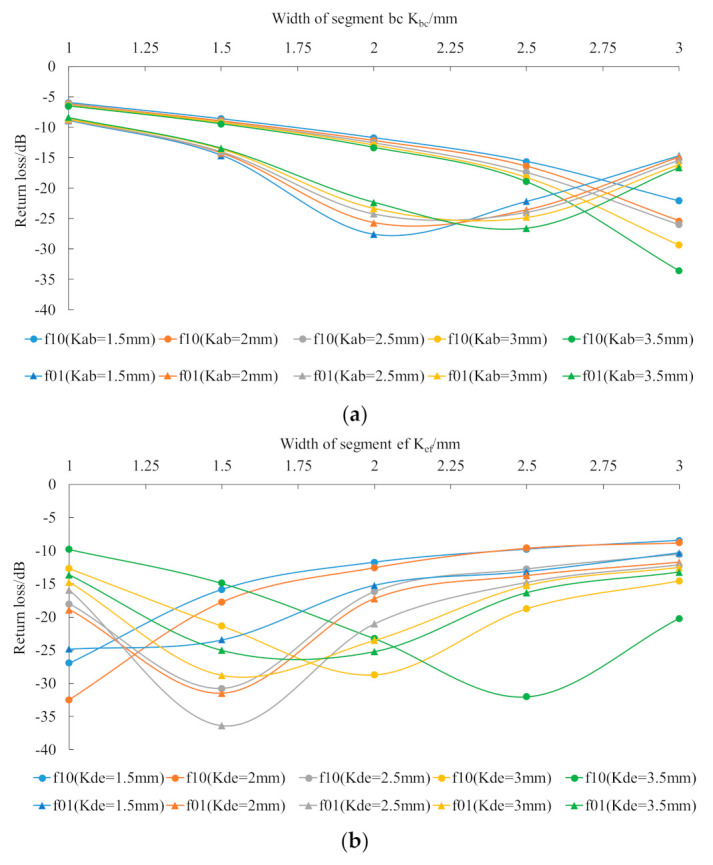
Simulation results for feeding line width optimization; (**a**) segment ab and segment bc: optimized simulation results; (**b**) segment de and segment ef: optimized simulation results.

**Figure 10 sensors-21-02766-f010:**
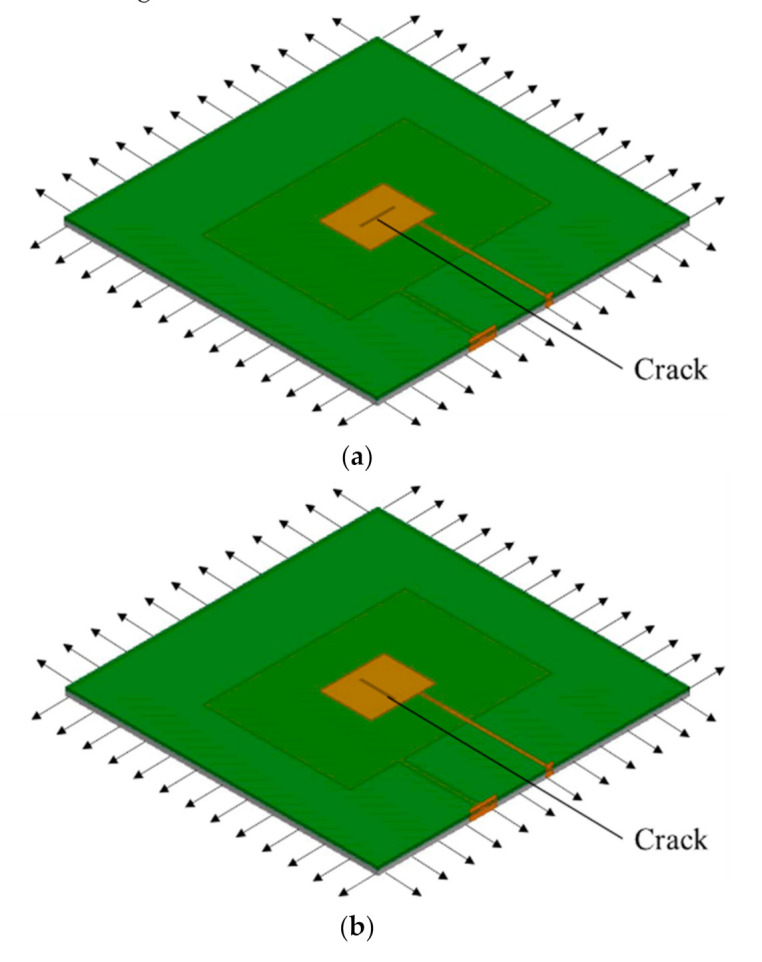
Schematic diagrams of crack propagation simulations: (**a**) schematic diagram of horizontal crack propagation simulation; (**b**) schematic diagram of vertical crack propagation simulation.

**Figure 11 sensors-21-02766-f011:**
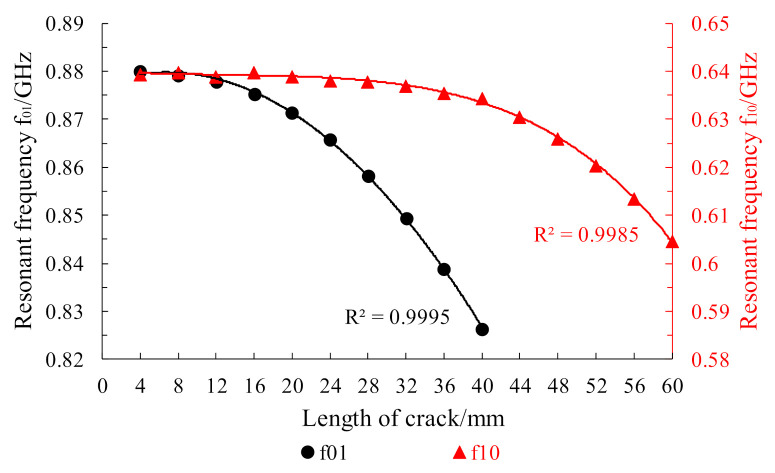
Changes in the resonant frequency of the underlying patch during crack propagation.

**Figure 12 sensors-21-02766-f012:**
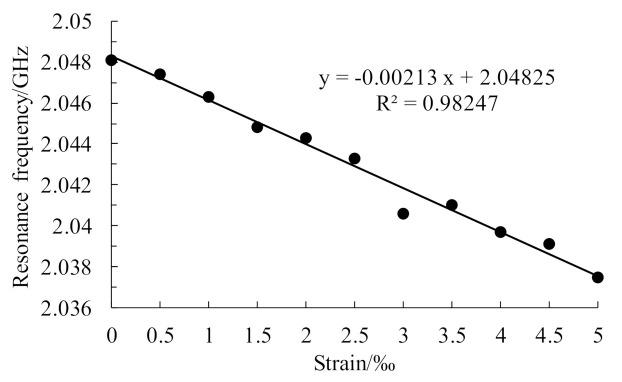
Variation in the resonant frequency of the upper patch with increasing strain (no crack-ing).

**Figure 13 sensors-21-02766-f013:**
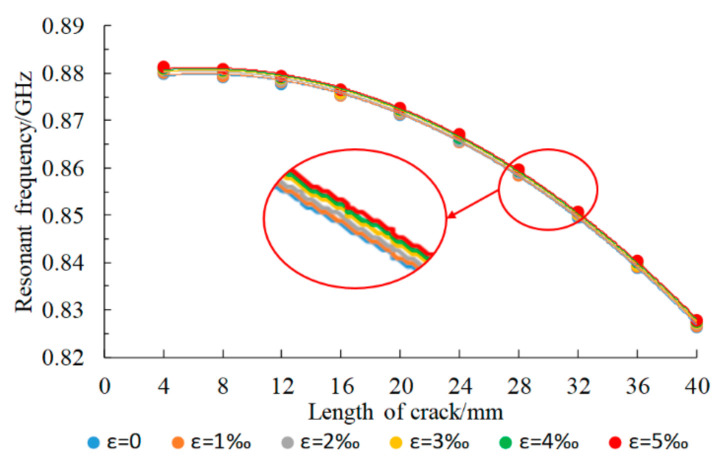
Variation of the resonant frequency of the underlying patch under different strain con-ditions.

**Figure 14 sensors-21-02766-f014:**
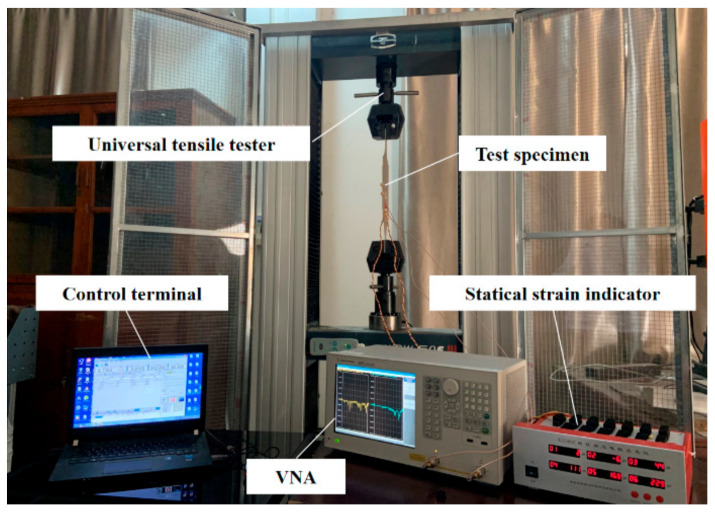
Test device.

**Figure 15 sensors-21-02766-f015:**
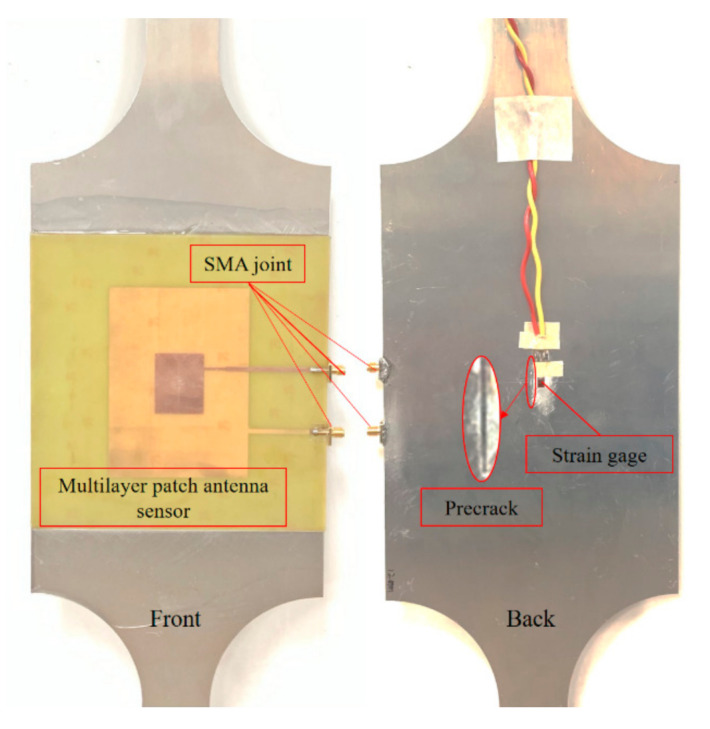
Arrangement of the test sample.

**Figure 16 sensors-21-02766-f016:**
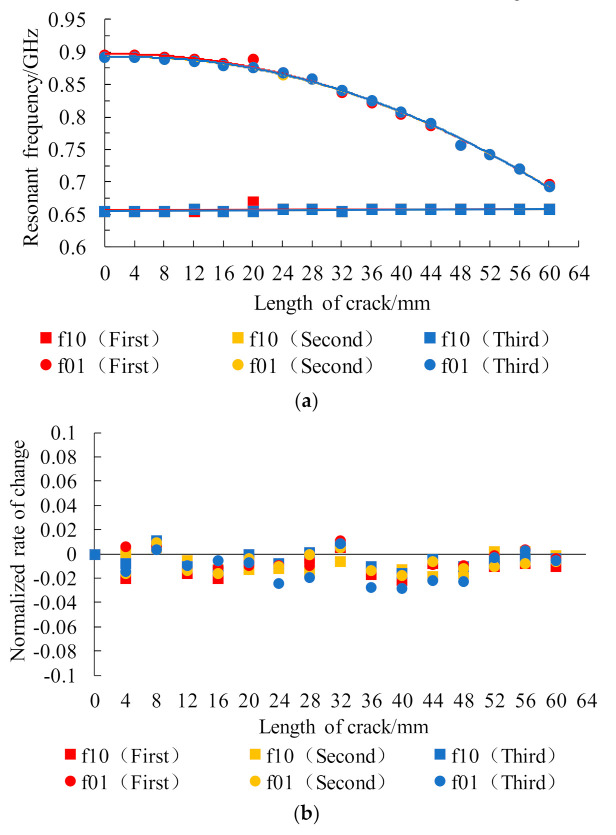
Resonant frequency changes of the multilayer patch antenna sensor during crack propa-gation: (**a**) resonant frequency of the underlying patch used for crack monitoring; (**b**) normalized rate of change of resonant frequency of the upper patch used for strain testing.

**Figure 17 sensors-21-02766-f017:**
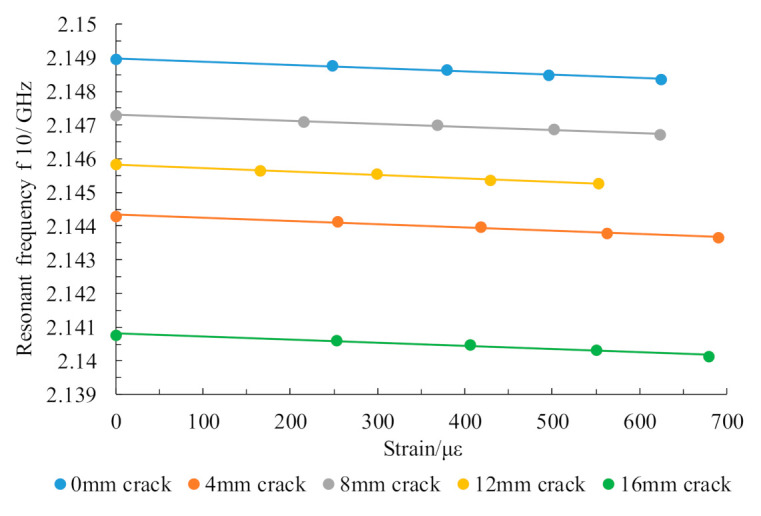
Relationship between resonant frequency and strain in the upper patch during crack propagation.

**Figure 18 sensors-21-02766-f018:**
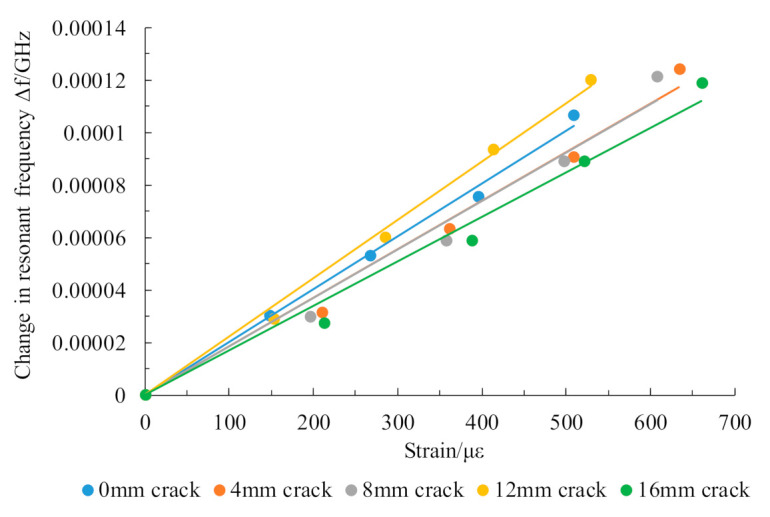
Relationship between resonant frequency and strain for the underlying patch during crack propagation.

**Figure 3 sensors-21-02766-f003:**
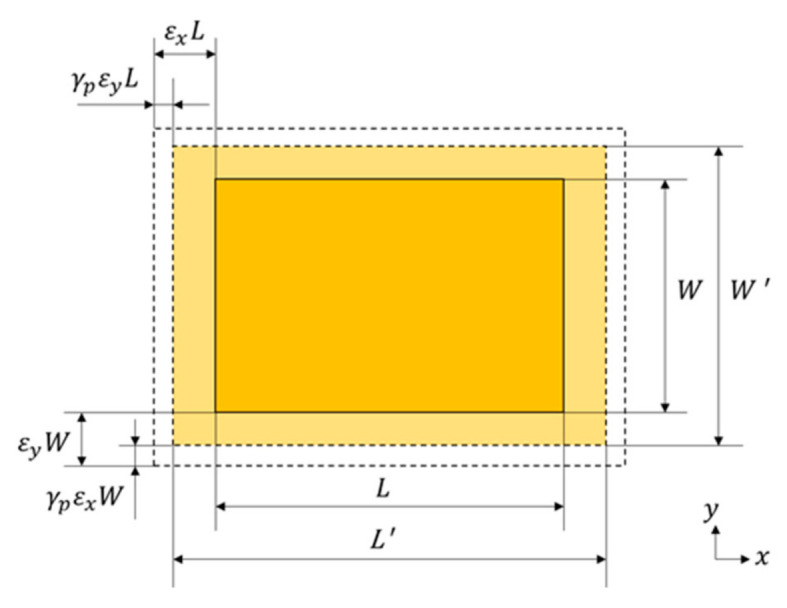
Deformation diagram of the patch under load.

**Figure 4 sensors-21-02766-f004:**
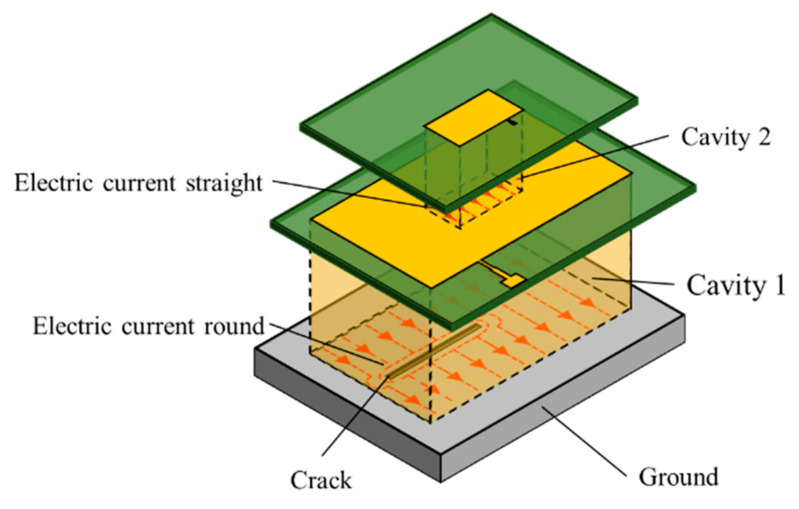
Schematic diagram of monitoring of strain and crack decoupling under multi-cavity model conditions.

**Table 1 sensors-21-02766-t001:** Design parameters of multilayer patch antenna sensor.

Patch Label	Patch Size (mm)	Resonant Frequency	Design Value (GHz)	Simulation Value (GHz)
1	Length $L_{1}$ 110	$f_{10}{}^{a}$	0.64	0.641
	Width $W_{1}$ 80	$f_{01}{}^{a}$	0.88	0.879
2	Length $L_{2}$ 35	$f_{10}{}^{b}$	2.05	2.048
	Width $W_{2}$ 28	$f_{01}{}^{b}$	2.54	2.542

**Table 2 sensors-21-02766-t002:** Design parameters of the multilayer patch antenna sensor.

Feeder Label	Length (mm)	Width (mm)
ab	5	2
bc	40	3
de	58.5	1.5
ef	12.5	2

**Table 3 sensors-21-02766-t003:** Relationship between strain and resonant frequency at different crack lengths.

Length of Crack (mm)	Resonant Frequencies at Different Strain Values (Parts) (GHz)					
	0.5‰	1‰	1.5‰	2‰	2.5‰	3‰
0	2.0474	2.0463	2.0448	2.0443	2.0433	2.0406
4	2.0467	2.0455	2.0453	2.0437	2.043	2.0419
8	2.047	2.0461	2.0453	2.0433	2.0428	2.0417
12	2.047	2.0462	2.0452	2.0439	2.0427	2.0415

**Table 4 sensors-21-02766-t004:** Mean values of the relative offsets at different strain states.

**Strain Size**	1‰	2‰	3‰	4‰	5‰
**The mean of the resonant frequency offset (MHz)**	0.19	0.55	0.89	1.15	1.41

## Data Availability

Not applicable.

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
