# Peer review of "Decoupled Monitoring Method for Strain and Cracks Based on Multilayer Patch Antenna Sensor"

_sensors, 2021, doi:10.3390/s21082766_

Round 1
Reviewer 1 Report
Authors, design a double-layer patch antenna in order to detect stress and cracks on cement. Manuscript is a bit lengthy and there are repetitions in the text like the operation of the double patch one for the cracks the other for the stress.
I am not convinced that stress tests can be successful when combined with the cracks, measurements are not very clear. On the other hand cracks seem to be found easier, when however are below the patch antenna. What happens when the crack is not below the antenna? In addition, I think that authors suppose that the metal is fully cracked. However what happens if there is crack on the metal but not from top to bottom? Details on the metal thickness under investigation are also missing (in relation with the previous remark).
Authors also design the λ/4 transformer carefully. However they assume that the metal is limited in size and therefore feeding is realisable and λ/4 line ends at the metal edge. What happens if the metal layer is big, but the line needs to be λ/4? How the antenna will be fed?
There are a couple of phrases that might not be commonly used or may be wrong:
Lines 249-252 do not make sense to me: What is the optimal state mentioned?
Also "echo loss" is an unknown term to me and is not used in antenna design / theory
"Feeders" is not an expression I am familiar with either. Do authors mean feeding lines?
I do not understand the unit khz/με. What is "με"
"Eccentric" feeding scheme is unusual term as well, I understand the meaning, but still authors should probably rephrase
Also authors need to explain why the patch antenna is surrounded by an electric and a magnetic wall and how this affects their analysis
"Benign conductors" is an unknown term to me
"Effective resonance" is another unknown term to me
The two modes (10 and 01) mentioned, is the result of the antenna different length and width. This is common in microstrip patches. The way it is presented, it is sees as it is the result of a meticulous design approach
I do not understand line 193. What is the contact and the non-contact type feeding?
Reviewer 2 Report
This is an interesting and well written paper, about (a) the design of a novel multilayer patch antenna sensor, and (b) the presentation of a novel method for decoupled monitoring of strain and cracks, which has to be published in the Sensors journal, after minor revision concerning the minor issues mentioned below.
Positive remarks about the paper:
- The paper’s subject is relevant to the journal and to the Section Fault Diagnosis & Sensors.
- The technical content is sound and well explained.
- It combines novel design with fabrication, as well as theoretical analysis and simulations with experiments.
- It proposes a smart decoupled monitoring method for the structural stress state and crack propagation.
- The authors analyze theoretically, simulate and verify experimentally the proposed strain and crack decoupling method, proving that simultaneous monitoring of the strain and crack information is feasible.
- It is well written in clear, idiomatic English.
- Previous related work is adequately referenced.
- It contains a lot of results, which might be really useful for researchers in the field, and especially those belonging to the Structural Sensors community.
- The results are promising for the health and safety monitoring of metal equipment structures.
- The designed multilayer patch antenna sensor has broad application prospects.
- The keywords accurately reflect the content.
Minor issues
- The authors should explain the origination of equations (2) and (3). If they come from literature, in which equations of which paper do they correspond? If not, then a schematic diagram of the strains generated along the patch length and width would be useful.
- The authors should provide comparisons of their results (calculated, predicted and measured) with those of other related works. How accurate are the novel sensor and the proposed method? The simultaneous monitoring of the strain and crack information is realized, but does the multilayer patch antenna sensor have advantages at all points, in comparison to the “simple” patch antennas used in other papers? One or more Figures and/or Tables containing comparisons with the corresponding literature are more than useful, when novel sensors and methods are studied.
I believe that the manuscript is suitable for publication in its present form, even though the authors should submit a slightly revised version of their manuscript, taking into consideration the above minor issues.
Thus, I propose a minor revision by (a) considering/answering above comments/requests, and (b) providing valid comparisons with other published results, in order to emphasize on the reasons making the addressed problem and the given solution important to the Structural Sensors researchers/readers.
Reviewer 3 Report
This paper presents a decoupled method to monitor strain and clacks. It is well organized and the claim has been supported by the experimental results. The literature has been reviewed enough. In the meantime, the reviewer would like to raise the following concerns to be considered.
- The variables in Figure 2 are not defined clearly.
- In Figure 15(a), why does the sample at x = 20 deviate from the curve unlike the others?
- Is it possible to draw the curves in Figure 10 together? As their scales are quite different while the shapes are similar, it is easy to misinterpret that the curves are analogue in every aspect.
- In Figure 11, how much are the deviations of the coefficients for the fitted line?
Reviewer 4 Report
The paper is dealing with monitoring of strain and cracks based on multilayer patch antenna sensors.
The paper is written in good style, using quite fine language. There are minor typos and formatting mistakes that should be removed from the final version.
Scientifically there is a good background description, simulation, sensor design explanation and experimental validation.
There are no significant remarks to be changed or improved.
Round 2
Reviewer 1 Report
Authors have addressed the points raised